# Health Support for At-Risk Older Adults during COVID-19

**DOI:** 10.3390/healthcare11131856

**Published:** 2023-06-26

**Authors:** Marian Ryan, Lisa M. Gibbs, Sonia R. Sehgal

**Affiliations:** 1Institute for Healthcare Advancement, La Habra, CA 90631, USA; 2Division of Geriatric Medicine and Gerontology, Department of Family Medicine, University of California, Irvine (UCI), Irvine, CA 92697, USA; lgibbs@hs.uci.edu (L.M.G.); ssehgal@hs.uci.edu (S.R.S.)

**Keywords:** health literacy, COVID-19 community education and behavioral mitigation, community-dwelling older adults, low-touch intervention, health equity

## Abstract

Older adults are highly susceptible to COVID-19 infection and at the highest risk for severe disease and death. Yet, older adults lacked access to accurate and easy-to-use COVID-19 information and support early in the pandemic. This prospective, experimental cohort study sought to examine whether older adults could be engaged during the pandemic through a community partner and if a low-touch intervention, designed with health literacy best practices, could positively impact COVID-19 knowledge, mitigation behaviors, telehealth/doctor visits, exercise, and loneliness. A senior resource kit was distributed to older adults sheltering at home through food assistance program agents from October 2020 to February 2021; the kit was developed using health literacy best practices. Simple random assignment was used to divide program participants into treatment and control groups. Both groups received senior kits, but the treatment group also received telephonic health coaching. The primary outcome was COVID-19 knowledge and mitigation behaviors as derived from self-reported surveys at baseline and after four months. Secondary outcomes included a telehealth or doctor visit, exercise frequency, and a loneliness score (3-Item Loneliness Scale). Health literacy was assessed using the BRIEF screening tool. Ninety-eight older adults consented to participate in the study and 87 completed the study (88.7% completion rate). Participants had moderate clinical risk, one-third preferred the Spanish language, and 52% were categorized as having inadequate or marginal health literacy. Significant changes were found for increasing COVID-19 mitigation behaviors and the frequency of exercise across the cohort, but not for COVID-19 knowledge, telehealth visits, or decreasing loneliness. Conclusions: Partnering with a trusted entity in the community is a feasible and important strategy to reach older adults during a lockdown and provide them with easy-to-read health information and resources. If the time horizon had been longer, improvements in other outcome variables may have been achieved.

## 1. Introduction

In early 2020, COVID-19 emerged as a serious disease caused by a novel virus that was spreading rapidly across the world and being compared with the influenza pandemic of 1918 that killed 500,000 Americans [1]. News coverage was incessant, frightening, confusing, and ever evolving. Public health information websites were written at literacy levels that exceeded recommendations for reaching the broadest number of people in the U.S. [2]. During this time of ever-evolving public health guidance on the COVID-19 pandemic, a consistent message from the Center for Disease Control (CDC) was, “those who are older and have a chronic medical condition are at increased risk for severe disease and death”. 

Older adults, the majority of whom live in the community, are at risk for falls, loneliness and depression, poor self-management of chronic conditions, limited technology skills, and limited health literacy before COVID-19. The pandemic exacerbated these risks for many older adults.

According to the CDC, 27.5% of older adults (65 years and older) reported falling at least once in the preceding year and 10.2% reported a fall-related injury. A higher percentage of older adults reporting no physical activity in the last 30 days were more likely to report a fall [3]. Research has also shown that physical inactivity weakens the immune system of older adults, making them more susceptible to infection [4]. Further, a lack of physical activity is important for older adults to maintain their independence and ability to perform activities of daily living and the authors suggest that health knowledge may encourage physical activity [5].

Older adults are at high risk for social isolation and at risk for increased psychological harm (depression and anxiety) from isolation [6,7]. Social isolation, loneliness, and social vulnerability are associated with morbidity and mortality that are comparable to established risk factors of smoking, alcohol, obesity, and frailty [8]. Physicians and health experts warned about the negative impact social isolation can have on older people’s immunity and mental health [7,9,10,11,12]. Early in the pandemic, social distancing became a troubling term for providers working with older adults as “social distancing challenged social health” and well-being, and physicality [13].

The highest prevalence rates of multiple chronic conditions are found among older adults and the number of chronic conditions increase with age [14]. Furthermore, older adults struggle more with managing their chronic conditions(s) [15]. They have difficulty with the mobile apps designed to help with self-management reminders, such as those for medications [16]. A Pew Internet Report in 2021 [17] found that the majority of older adults have “lower-tech readiness”, defined as needing someone else to help them use a computer or smartphone and/or no to low confidence in their ability to do what they need to do online. 

Only 3% of older adults, 65 years and older, were proficient in health literacy skills according to the only national study of health literacy conducted in the United States; 29% had below-basic health literacy [18]. Personal health literacy as defined by Healthy People 2030 is the degree to which individuals have the ability to find, understand, and use health information to inform health-related decisions for themselves and others. Many older adults experience age-related changes such as vision, hearing, cognition, and physical functioning [19] in addition to limited health literacy. The progression of frailty in community-dwelling older residents can be predicted by limited health literacy [20]. Moreover, the highest levels of health literacy were needed to make good personal decisions for oneself and one’s family given the scientific uncertainty of the novel coronavirus and the inconsistency in public health messages [21,22]. 

Many healthcare professionals were worried about the health and well-being of older, community-dwelling adults as lockdowns were implemented to limit the spread of COVID-19. With the closure of medical offices and the rapid pivot to telehealth services, many older adults would be challenged to access the care they needed [23,24,25]. 

This feasibility study aimed to test a new approach to help older adults understand public health guidance, stay safe, speak with their doctor, engage in tele-health visits, exercise at home, use technology to connect with loved ones, and engage in active living during “stay-at-home” orders. 

The objectives of this study were as follows:Examine the feasibility of “low-touch” intervention to engage older adults through a trusted community partner and facilitate access to reliable COVID-19 information.Analyze improvement rates in COVID-19 knowledge and mitigation behaviors, telehealth visits, loneliness, and at-home exercise as a means of fall prevention.Assess implementation factors that supported and hindered engagement.Assess the seniors’ perception of the intervention, resources, and the pandemic.

## 2. Materials and Methods

### 2.1. Participants

Residents of La Habra, California, participating in the senior food assistance programs were invited to participate in the study over a four-month period. During a routine meal delivery service, food program drivers distributed the Senior Health Resource Kit with the study invitation and informed the clients that someone would phone them within two days. Researchers phoned all clients to describe the study, explain the contents of the kit, answer questions, and obtain consent. Exclusion criteria were age under 60 years and a language preference other than English or Spanish. An IRB review was conducted by the University of California, Irvine, and internally approved.

Over a rolling 16-week period (October 2020 through February 2021), 203 Senior Heath Resource Kits were distributed by the food program drivers; 98 seniors were successfully enrolled for a 48.2% engagement rate. Over the 4-month study period, 11 people were lost to follow-up and 87 completed the study, for an 88.8% completion rate. Data were analyzed for the 87 participants completing the study. The 11 non-completers were those who could not be reached at the end of the fourth month to complete the end-of-study survey. Bivariate analyses were conducted on all baseline parameters between the 11 who were lost to follow-up and the 87 who completed both the baseline (enrollment) and post-study surveys; no differences were observed. As this study was conducted early in the pandemic, we could not visit study participants to collect end-of-study data if they did not respond to outreach phone calls. We called four times and mailed two reminders.

More than half of the study participants were Latino (52%) and 36% preferred the Spanish language. Nearly a third (32%) had an education less than 8th grade. Age was evenly distributed across five age categories with 18.4% aged 60 to 64 years, 29.7% aged 65 to 69, 20.8% aged 70 to 74, 20.7% aged 75 to 79, and 18.4% aged 80 years or older. More than half the study population had either inadequate or marginal health literacy; 29% had inadequate health literacy; and 23% had marginal health literacy. Nearly 29% had difficulty paying for prescription medications and 46% did not have internet access.

This group of community-dwelling older adults reported a moderate risk profile: 2.8 chronic conditions on average (33% had diabetes); 23% taking more than six prescription medications; and in the last year, 38% sustained a fall, 31% had at least one ED visit, and 22% at least one hospital stay. Baseline results were equivalent between treatment and control groups on self-reported variables including clinical risk, health literacy, internet use, and self-efficacy (Table 1. Demographic and clinical descriptive statistics at baseline).

### 2.2. Study Design 

This prospective, experimental study was designed with rolling enrollment and following consent, random assignment to the control, or enhanced intervention. The only exception to this assignment process was for spouses or partners living in the same house. In this situation, both people were assigned to the group assignment of the first person. This was performed to protect against the potential “spill-over” effects of treatment on the person assigned to the control group. 

The baseline survey was developed incorporating validated screening questions where possible. It included questions about their doctor, doctor visits, health status [26], hospital stays and ED visits, chronic conditions, number of medications and medication access, COVID-19 (testing, knowledge, and mitigation behaviors), internet and cellphone use, self-efficacy [27], exercise frequency, loneliness [28], health literacy [29,30], and general demographics. 

A single health status question was used, shown to be highly correlated with the SF-12 measurement for self-reported physical quality of life [26]. Access to medication was collected with 2 questions, “Did you ever have problems getting medicine during COVID-19? In the last 6 months, did you ever run out of your medicine and stop taking it”? 

Health literacy was assessed using the BRIEF Health Literacy Screening tool [29,30] as it has been validated and tested over the phone with older adults. The four questions include the following:How confident are you filling out medical forms by yourself?How often do you have someone help you read medical materials?How often do you have problems learning about your medical condition because of difficulty understanding written information?How often do you have a problem understanding what is told to you about your medical condition?

Response variables to the BRIEF are scored on a 5-point Likert scale (Q-1, response options range from not at all to very sure; Q-2 to Q-4, response options range from always to never). The BRIEF screen categorizes health literacy as inadequate health literacy (4 to 12 points); marginal health literacy (13 to 16 points); and adequate health literacy (17 to 20 points). 

Study research assistants, trained in research ethics, the consent process, and survey protocols, administered the baseline survey. Study enrollment was conducted between October 2020 and February 2021 and follow-up surveys were completed from February 2021 to June 2021, four months after the administration of the baseline survey. Surveys were completed over the phone and USD 10 gift cards were given for the completion of each survey. 

Survey protocols (scripted questions and prompts) were written to ensure a standardized process for the implementation of the surveys. The food assistance program staff were also provided information about the study and the benefits of participation.

The study was launched during the period of the “stay-at-home” order in California whereby older adults were instructed to remain in their homes as much as possible to keep themselves safe. Recruitment for the study was possible only through a partnership with the La Habra senior food assistance programs, trusted by the residents. 

### 2.3. Data Collection

Data were primarily collected from participants’ completed baseline and post-study surveys. The survey was developed and iterated following internal, informal field testing by researchers. Many of the questions were adapted from questions in Medicare and Medi-Medi Health Risk Assessment surveys used in the County. Medical history and clinical questions were included to establish a study cohort risk profile and identify at-risk older adults in need of health care and services. As previously discussed, research assistants were trained in the administration of the survey and using scripted protocols for questions and prompts if needed. Three of the four research assistants were proficient in both English and Spanish languages. Most of the questions in the survey came from other validated surveys or surveys in current use by health plans that are available in English, Spanish, and a variety of other languages and are administered telephonically.

The key outcome variables were COVID-19 mitigation behavior and COVID-19 knowledge. These questions were developed by aligning with stay-at-home state and public health orders. The first three COVID-19 mitigation behaviors included the frequency of leaving the home (older adults were to stay at home as much as possible), visits by groups of friends/families (older adults were not to have groups of people in their homes), and attending large gatherings (no one was supposed to attend large gatherings)—response options included never to weekly (scored 5 to 1; reverse scored as these are undesirable behaviors). Wearing masks (first mandate of the public health order) when going outside was measured with response options of never to always (scored 1 to 5), and the likelihood of going without or delaying urgent care because of COVID-19 was measured with response options of very unlikely to very likely (scored 5 to 1). Early in the pandemic, many physicians were concerned that older adults may be avoiding the ED in serious situations due to fear; this has been confirmed [31,32]. Summing the scores for each question led to a composite score, ranging from 5 to 25.

COVID-19 knowledge was assessed by three questions believed by the authors to be important to keeping oneself safe and healthy: (1) Can people have COVID-19 and not know it? (2) Is it safe to go to an event where many people are present, as long as, they are 6 feet apart? (3) Is it safe to go to the hospital if you are really sick?

One point was assigned for each correct response yielding a COVID knowledge composite score of 0 to 3. 

Other study outcomes included improving access to telehealth, increasing exercise, and decreasing loneliness. 

Access to telehealth visits was assessed by two survey questions: (1) In the last 6 months, did you have a telehealth visit with your doctor? (2) In the last 6 months, did you see your doctor in-person? The post-survey timeframe for these questions were modified to “In the last 4 months, did you…”. These questions were converted to a single binary variable (1/0) with a 1 assigned if they had a visit with their doctor.

Exercise frequency was measured at baseline and post study with a single question, How many days a week are you doing some kind of exercise now? Response options included: No days, 1 day, 2 to 3 days, 4 days, 5 days or more, scored as 0, 1, 2, 3, and 4, respectively.

Loneliness was measured using the shortened version of the UCLA Loneliness Index [28]. The survey consists of three questions: (1) How often do you not have someone to talk with? (2) How often do you feel left out? (3) How often do you feel disconnected from others? (Response options for all three questions are as follows: hardly ever, some of the time, often, scored as 1 to 3). The index ranges from 3 to 9; greater than or equal to 6 indicates high risk for loneliness—this short version received an acceptable reliability score of 0.72. 

### 2.4. Intervention Group

All potential study enrollees received a Senior Health Resource Kit—the planned, low-touch intervention to engage older adults. The resources were placed within a box with handles, adorned with photos of happy older adults. These were delivered by food program drivers at the time of the meal deliveries to their homes.

The Senior Health Resource Kit included a flyer about the study with a cover letter, a health book for older adults (*What To Do For Senior Health*), a COVID-19 flyer with clear information on staying safe and resource numbers (warm phone line for depression or anxiety, health department information line, and several reliable websites), a home fall assessment booklet for fall prevention, a fall prevention exercise DVD and corresponding YouTube URL, a consent form, and a copy of the survey. The kit also included helpful items—an LED light, magnifying bookmark, and eyeglass cleaner and cloth. All written materials were produced using health literacy best practices. All the resources in the senior kit had been translated into Spanish, field-tested, and revised prior to the study.

In addition to the Senior Health Resource Kit, the intervention group received calls over the four months from a health coach for ongoing support. Two bilingual health coaches were trained in the intervention protocols (developed for this study) and provided training and resource binders. 

Within two days of completing the baseline survey, a health coach called those in the intervention group to answer questions and provide support. The initial priority for the health coach was to address any risks identified on the baseline survey. Priority support included facilitating telehealth visits with their doctors and prescription medication receipts if needed. After these risks were resolved, or if the older adult did not have identified risks, the support coach then reviewed all the resources in the kit and how to stay well during COVID-19. They facilitated computer and smartphone training if needed, explained the home fall assessment, and problem-solved with the study participant on remediation strategies.

The support coach reviewed basic fall reduction exercises that can be performed in the home, and facilitated access to the fall prevention exercise video where each exercise is demonstrated. On every coach call, the exercise was assessed, and they were encouraged to use the video exercises to reduce fall risk and maintain physical conditioning. Exercises used were evidence-based and demonstrated to reduce falls [33,34]. 

### 2.5. Control Group 

The control group received the same Senior Health Resource Kit as the intervention group—the planned, low-touch intervention to engage older adults. 

After administering the baseline survey to the control group participants, the research assistants provided accurate information related to COVID-19 knowledge and mitigation behaviors and answered any additional questions about the kit contents. They encouraged the study participants to stay in contact with their doctors, continue to take their medications, read the parts of the senior health book about staying healthy, exercise at home using the DVD or YouTube video, reach out to their family and friends using the phone or computer, and stay safe by following the recommended COVID-19 mitigation behaviors. 

The research assistants informed control participants that they would be called again in four months to complete the end-of-study survey. 

### 2.6. Outcome Measures

Primary outcome measures were COVID-19 mitigation behaviors and COVID-19 knowledge. Primary goals of this study were to improve both personal mitigation and COVID knowledge to keep the older residents as safe as possible. 

The COVID-19 mitigation behaviors were assessed as an interval composite variable where the higher the score, the more the person was fully following recommended mitigation behaviors. Given the sample size and scale range of this variable, a parametric test for this outcome measure may be used [35,36]. In the final analysis, a mixed linear model with repeated measures of treatment and time on the COVID mitigation behaviors score with a random intercept for pre/post measures by treatment group was estimated. 

COVID-19 knowledge was collected with three questions. For analysis, this variable was converted to a binary variable (1/0) with a 1 assigned if all three questions were answered correctly. As it was expected that COVID-19 knowledge would impact behaviors, it was important to determine baseline and post COVID knowledge.

Secondary outcome measures included loneliness, telehealth access, and exercise. The 3-Item Loneliness Index was converted to a binary variable (1/0) with a 1 assigned for at-risk for loneliness if the total score was 6 or more for the analysis. The 3-Item Loneliness Index represents a screener for identifying people with high levels of loneliness for whom interventions would be appropriate. This study engaged with a group of older adults sheltering at home and often with little to no contact with others. It was believed participation in the study would decrease their risk for loneliness. 

Exercise frequency was used to examine changes in exercise during the study. At-home exercise video (and private URL) was included in the senior kit and participants were encouraged to perform some of these every day. This component was designed to prevent avoidable falls in the home and maintain function during this time of staying at home. Exercise frequency was analyzed as an ordered, multinomial variable. 

### 2.7. Statistical Analysis

Our study population was described using proportions for categorical variables and means with standard deviations for continuous variables as a cohort and by group assignment (“treatment” versus “control”). All outcome measures at baseline were compared by group assignment. Bivariate statistics were used to estimate a change in rate for COVID-19 mitigation behaviors and knowledge, loneliness, telehealth, and exercise. The paired *t*-test was used for COVID-19 mitigation behaviors (composite was treated as interval data) and the chi square (or Fisher’s exact test for small cell size) statistic was used for categorial variables as appropriate for significance testing of differences between pre and post measurements. 

The main objective of the study was to determine if a “low-touch” intervention (Senior Resource Kit containing resources that were easy-to-read and easy-to-use) could be effective in improving COVID-19 mitigation behaviors and knowledge to keep older adults safe during the early pandemic and if health coaching would further enhance the impact of receiving the Senior Resource Kit. For these reasons, a Difference-in-Differences method was used to analyze changes in pre- and post-study outcomes between treatment and control groups. The benefit of such a model is that it accounts for secular changes occurring in the environment exposing both study groups to parallel trends across time. In our study, the cohort was fixed over time controlling for measured and unmeasured confounding. Generalized linear models were used to generate the “difference-in-differences” estimates for all outcome measures using the appropriate model specification for the variable type (continuous, binary, or ordinal). A difference-in-differences (D-I-D) estimate is the difference between the difference in treatment post- and pre-parameter estimates, and the difference in control post- and pre-parameter estimates, sometimes referred to as a controlled before and after study as participants serve as their own controls. The D-I-D estimates the between-group cross-sectional differences and the within-group time series differences to measure treatment effects [37,38,39,40,41,42]. We tested the dummy variable for inadequate health literacy as a fixed effect in all outcome models as it is a key factor in this study. All statistical tests were two-tailed with a critical alpha equal to 0.05. All quantitative analyses were performed using SAS proprietary software 9.4 and analytical products SAS/STAT 15.1 (SAS Institute, Inc., Cary, NC, USA).

Importantly, the minimum requirements and key assumptions for a D-I-D model were satisfied [43]. The D-I-D approach requires the collection of outcome data on an exposed or treated group and a group not exposed to the intervention (control) during at least one time period before the exposure or intervention and at least one time period after the intervention. The allocation of treatment cannot be determined by the outcome; our intervention was by study design and not allocated by the outcome. The treatment and control groups can be assumed to have parallel trends in outcome. As only baseline and post-4-month measurements were collected, the parallel trend prior to the intervention cannot be directly measured. However, the short timeframe between pre-intervention and post-intervention data collection and the randomization of the participants to the treatment and control groups support this assumption holds. The composition of the treatment and control groups was stable over time and no differences were observed at baseline between the groups, nor were any differences found between the 87 older adults completing the study and the 11 study participants who could not be contacted for post-data collection. Repeat measures were collected on the same individuals at both time periods and the mixed procedure model with random effects accounted for the correlation between these measures for the same individual. There were no known spillover effects during the study. This study took place at the height of the COVID-19 pandemic and study participants in the treatment and control groups lived in the same city and were exposed to the same public health orders, COVID-19 infection rates, and access to preventive COVID-19 education and resources. Additionally, simple random assignment procedures kept two older adults living in the same household in the same group; older adults were assigned to the random assignment of the first person in the household.

A separate difference-in-differences regression model was estimated for each study outcome. All models compared baseline data collected between October 2020 and 15 February 2021 with post-study data collected from 16 February 2021 to 31 May 2021.

For the primary outcome of interest, COVID-19 mitigation behavior, a mixed linear model with repeated measures of treatment and time on the COVID mitigation behavior score with a random intercept for pre/post measures by treatment group was used to generate the difference-in-differences estimate and the fixed-effects parameter estimates. The estimation method was Restricted Maximum Likelihood.

The COVID knowledge, exercise frequency, loneliness index, and no telehealth (or other doctor) visit were modeled using generalized estimating equations (GEE). This permits an extension of the traditional linear model by allowing the mean of a population to depend on a linear predictor through a non-linear, link function [39,41,44]. For each outcome, GEE was used to generate a population-level marginal difference-in-differences estimate. The model regressed the interaction between post and treatment on the dependent variable with a specified distribution and link function (based upon the dependent variable characteristics).

Three focus groups were held (two in English and one in Spanish) and five individual interviews (four in English and one in Spanish). Focus groups and interviews were used to assess the study participants’ perceptions of the Senior Kit as an intervention, and how they were coping during the pandemic. Semi-structured focus group and interview guides were used and translated into Spanish for the Spanish language group. Focus groups and interviews were taped, and the taped Spanish focus group was translated into English for coding and analysis. Plain-language instructions for using Zoom were developed and mailed to those participating. Despite calling participants to review by connecting to Zoom, some participated by phone only. Thematic analysis was used to identify, analyze, and report patterns in the data [45].

## 3. Results

### 3.1. Primary Outcomes

At baseline measurement, COVID-19 mitigation behavior (composite score) was statistically higher for those adults assigned to enhanced treatment versus those assigned to the control. The independent samples *t*-test statistic estimated a −1.43-point difference in the composite scale between participants assigned to the control compared with those assigned to treatment (*t* value of −2.31, *p* = −0.024). (Table 2. Outcome measurements at baseline).

At end-of-study, a change in the COVID-19 mitigation behavior score was evaluated using a “difference-in-differences” method. In identifying potential model covariates, Pearson correlation coefficients were estimated for the association between the COVID mitigation behavior score at baseline and all collected demographic and clinical data at baseline. Variables with a Pearson correlation coefficient <0.18 were tested in the model; comparisons of best-fit model statistics, inadequate health literacy, ED visit in the last year, hospital stay in the last year, and Spanish language were retained in the final model.

The D-I-D estimate was significant (mean estimate, −1.5628, *t* Value = −2.07, *p* > |*t*|, 0.0417). As the estimate was negative, the slope of the treatment group at the end of the study was less than that of the control group and this difference is statistically significant at the 0.05 level. The Least Squares Means (rounded to 1 decimal point for simplicity of math) estimated Treatment group post mean = 20.5, pre-mean = 18.9 (1.6 difference); Control group post mean = 20.8, pre-mean = 17.6 (3.2 difference); difference in differences = 1.6 − 3.2 = −1.6.

The fixed-effects parameter estimate “Post” for COVID mitigation behavior score was significant and positive, indicating that mitigation behaviors in the study cohort increased during the post period—estimate, 3.213; *t* value, 6.27; *p* < 0.0001. Fixed-effects parameter estimates for ED visit in the last year and Spanish language were significant, whereas those for inadequate health literacy and hospital stay in the last year were not significant. (Table 3: Results—difference-in-differences estimates for study outcomes). 

At baseline, no differences between treatment and control groups were observed in the other outcome variables—COVID-19 knowledge, no telehealth (or other doctor) visit, exercise frequency, and loneliness index (Table 2). Overall, across the study cohort, 52.9% had good COVID knowledge (all three questions correct), 37.9% were exercising 5 or more days per week, more than a third (35.6%) screened for loneliness risk, and 13.8% had no telehealth or other doctor contact in the last 6 months.

No change was observed in COVID-19 knowledge at the end of the study, modeled using generalized estimating equations (GEEs). In the case of COVID-19 knowledge, GEE was used to generate a population-level marginal difference-in-differences estimate. The model regressed the interaction between post and treatment on the binary dependent variable (all three questions correct) with a specified binomial distribution and identity link function. Inadequate health literacy and Spanish language covariates were tested in the model, but the model performed better without them. The D-I-D estimate was not significant, and the independent post and treatment parameter estimates were not significant (Table 3: Difference-in-differences estimates for study outcomes).

### 3.2. Secondary Outcomes

A significant difference-in-differences estimate was not observed for the frequency of exercise at the end of the study. A generalized estimating equations model with a specified multinomial distribution and cumulative log link was used to estimate a population-level marginal D-I-D model. The D-I-D estimate was not significant; however, the estimate for “Post” was significant—the parameter estimate was 0.698, the *t* value was 3.76, and *p* = 0.0002. Although the difference between the within-group post/pre differences and cross-group differences was not significant, exercise did increase across the cohort over the post period. 

Loneliness was analyzed as a binary outcome, high risk for loneliness, and as a multinomial outcome, Loneliness Index. The GEE model was first specified with a binomial distribution and identity link converting the dependent variable to a binary value (1/0), indicating a high risk for loneliness. All estimated parameters in this model were insignificant. Given the loss of data in converting the 3- to 9-point scale to a binary value, a GEE model was repeated with the full loneliness scale as the dependent variable and specified multinomial distribution and cumulative logit link. Although the model was superior, the D-I-D estimate and all other parameter estimates were not significant.

Participants with no doctor telehealth visit or other contact with their doctor did not decrease significantly over the study. A GEE model specified with a dependent variable of “No telehealth visit” and specified with a binomial distribution with an identity link function was estimated. Insignificant parameter estimates were observed. 

Nearly 60% of the study population used the home assessment book in the Senior Resource Kit to identify potential fall risks in their homes; 63% of these older adults fixed potential fall risks or had them fixed, and 12% reported using the assessment and finding no potential fall risks. 

Due to the timing of the study, vaccination rates were not planned as an outcome. However, they were collected at the end of the study. There was a significant difference in COVID-19 vaccination of 85.0% (“treatment”) versus 63.8% (“control”), *p* = 0.026 (chi-sq.). Overall, 73.5% (64) of study participants had received COVID-19 vaccinations by their end-of-study surveys; seven had not had their opportunity to get the vaccine yet but intended to get it. Among this group of older adults, most received information about COVID-19 from the TV (66.7%); only 8.3% used the internet regularly.

### 3.3. Implementation Factors That Supported Engagement

We explored covariates to identify possible reasons for our lower-than-anticipated engagement rate, 48.2%; 98 older adults were consented, surveyed, and joined the study at baseline of the 203 kits distributed. No statistically significant difference was found in joining the study by language or by the type of food assistance program. Differences in the outcome of outreach calls were found by language. There was a higher rate of active refusals by English-language older adults compared with Spanish-language older adults (24.0% vs. 12.2%, English vs. Spanish, respectively, *p* < 0.0001 [Chi-sq]). Unable to Contact or Reach outcomes were higher among older adults identified as Spanish language speakers (24.8% vs. 41.5%, English vs. Spanish, respectively, *p* < 0.0001 [Chi-sq]). Research analysts made multiple phone calls to reach potential participants after kits were dropped off by food program drivers. On average, 2.3 outbound calls (range, 1 to 4) were made for each successful contact. A reminder postcard and two letters were mailed after the fourth unsuccessful phone call.

The factors that facilitated implementation and engagement with old older adults sheltering at home included the following: established partnerships with all the senior food programs with home delivery (Meals on Wheels) or senior centers that pivoted to the home delivery of meals due to COVID-19 closures; quickly facilitated data exchange agreements and securing only the minimum amount of client data needed (name, language, and telephone); Institute for Healthcare Advancement (IHA) had developed a Senior Health Resource Kit that had been previously evaluated before the COVID-19 pandemic; respect for UCI Irvine and knowledge of IHA in the community facilitated a trusted partnership; rolling study enrollment allowed research analysts and health coaches to manage the workload without time delays in outreach.

### 3.4. Seniors’ Perceptions about the Intervention, Resources, and the Pandemic

We held three focus groups (two in English and one in Spanish) and conducted five interviews at the end of the study using Zoom and telephones. The transcripts were reviewed after translation and common themes were identified. Overall, there was a high level of stress expressed by almost everyone during the pandemic—fear, anxiety, constant worry, grief, and loss being the greatest contributors of stress. Almost everyone had known someone who had died during the pandemic, and they expressed the difficulty getting through it without the typical support of being surrounded by family and friends at a time of grief. They expressed being mentally “down” and very much alone.

Participating in the study made them feel safe and supported—“like someone cares”, and that “Someone is listening to us”. Everyone showed appreciation for the Senior Kit itself and valued the information. Many placed the box in a central spot in their homes so it could be easily accessed. They also added medical information to the box. Many shared that the health book was easy to read and visually comfortable. They all said they learned something they had not known about aging and staying healthy. Many of the older ones in the group expressed the wish to have received this book about 10 or 15 years earlier. Everyone liked the magnifying bookmark and LED light. “Having this information helped my quality of life—it’s better now that I know who to call—everything is tenfold better!” one respondent shared.

## 4. Discussion

In this feasibility study of community-dwelling older adults residing under stay-at-home orders, a “low-touch” intervention, a Senior Resource Kit with engaging, easy-to-read materials on staying safe and healthy during COVID-19 improved mitigation behaviors recommended by public health, and the state and local authorities (<0.0001). Study participants were randomized into control and treatment groups; however, the entire cohort received the Senior Resource Kit as the “low-touch” intervention, and the treatment group additionally received health coaching support calls. Contrary to our hypothesis, the health coaching support calls did not enhance this “low-touch” intervention (written materials and DVD) and increase our mitigation behavior score beyond the improvement observed in the control group. The difference-in-differences estimate was negative (−1.563) and significant (<0.42). Although both study groups increased their COVID mitigation behavior scores, the difference between each study group’s mean score difference (post-pre) was larger for the control group, not the treatment group. Notably, the control group’s least squares mean mitigation behavior score at baseline (17.6) was much lower as compared with the treatment group (18.9). The post-mitigation behavior scores were closer at the end of the study, but the control group was slightly higher (20.8 versus 20.5). No differences were found when comparing the demographic and clinical data between treatment and control groups—even for variables one might anticipate influencing COVID mitigation behavior.

This unexpected finding could be that the health support calls focused on other health concerns—the study protocol was for the health coach to first attend to any risk responses from the participant at baseline. Another possibility is the plain language materials regarding COVID-19 and how to keep oneself safe improved any confusion an older adult may have had about public health recommendations [46]. As these materials were engaging, easy-to-read and understandable, and delivered by a trusted person, the strength of the Senior Resource Kit alone increased the mitigation behavior of both groups [47,48]. Perhaps older adults assigned to the control group reviewed the materials more often because they did not receive coach calls.

There was no observed change in the difference-in-differences estimate for COVID-19 knowledge and no increase across the cohort. The COVID-19 knowledge variable was assessed as a binary variable—responding to all questions correctly because the information contained in these questions were required to follow mitigation behaviors. The non-significant negative estimate for treatment (health support calls) may suggest COVID-19 knowledge was not a focus of the calls.

There were no significant findings on our secondary outcomes for the difference-in-differences estimate in support of our hypothesis that the difference between the treatment (post-pre rates) and control (post-pre rates) would be positive for increase exercise, decrease loneliness, and decrease the number of older adults without a telehealth or other doctor contact. Very unexpectedly, only one of our secondary outcomes had any significant improvement across the cohort over the post period. An increase in exercise to preserve physical function and support well-being over the post period of the study was significant (*p* = 0.0002). It could be that both groups were supported in exercise that could be done at home and it made them feel better to exercise.

Insignificant findings for decreasing loneliness across the cohort may be the result of the chosen screening instrument for a “lock-down” situation. The instrument was chosen because it was brief, specific to loneliness, and had been validated. It was expected that the health support calls would decrease loneliness through these regular contacts. The non-significant treatment parameter estimate (−0.594, *p* = 0.105) may suggest that the calls did decrease loneliness of the older adults in the treatment group, but not quite to a statistically significant level. The insignificant finding in the lack of telehealth (or other doctor contact) outcome was not surprising. Only 13.8% of the study cohort (12 older adults) had not had a telehealth visit or other contact with their doctor at baseline. These baseline results were unexpected as this was an expressed concern from physicians. At the end of the study, four of these twelve older adults did have a visit with their doctors, a clinically although statistically non-significant meaningful result.

Studies of effective health communication strategies in an emergent situation are limited [47,49]. The last time a public health emergency played out this extensively was over one hundred years ago, but the few studies shown indicate that a trusted messenger is critical and plain-language information hand-delivered to residents is needed to meet the needs of vulnerable populations [47,48,49].

Our qualitative findings from focus groups and interviews were consistent with research during the timeframe [50,51,52]. The COVID-19 stay-at-home orders were challenging times for older adults and innovative ways to communicate with them are critical [53]. It resulted in a negative impact on their health, mental health, and physical health and the resulting social isolation and decreased physical activity further threatened their health [54]. An intervention to encourage at-home exercise during this period appears prudent.

This study had several limitations and was intended as a pilot study to test a new approach (trusted and accessible messenger and easy-to-read and act-upon materials to deliver critical health information and support to older adults during the pandemic lockdown. The study population was restricted to a single location although representative of older adult risk in many other areas—culturally diverse, eligible for the senior food assistance program, and of moderate clinical risk. The COVID-19 knowledge and mitigation behavior questions were limited to minimize survey burden and the desire to gather medical risk data to accurately profile this community population. The engagement rate was much lower than anticipated at 48% and no information could be collected about those older adults who received the Senior Health Resource Kit (the “low-touch” intervention) but failed to join the study. The study was conducted during a pandemic that included challenges for older adults, especially in underrepresented communities who were hardest hit [55]. The chaos during this time may have overshadowed those who would have participated in the study otherwise. Also, the survey itself comprised questions from multiple validated surveys; however, the overall survey was not validated.

The failure to find significant differences in many of the outcome measures may be related to the short length of follow-up (four months) and the failure to recruit as many participants as intended during the pandemic.

## 5. Conclusions

Although this study was conducted as a feasibility study, it has important implications. The study did demonstrate the feasibility of the approach—the engagement of older adults through a trusted community partner (in our case, senior food program drivers), and delivery of a “low-touch” intervention with accessible health information from a trusted person. Although we failed to find a positive impact from the enhancement of health support calls (using a difference-in-differences estimation), we did observe an increase in COVID-19 mitigation behavior across the overall cohort (<0.0001). We also observed an increase in exercise across the study cohort (*p* = 0.0002) that has the potential to reduce avoidable falls, improve immune response, promote feelings of well-being, and sustain independence among older adults.

A trusted community partner was an essential implementation factor, but the timeline for recruitment and the study duration should have been extended. The older adults enrolled in the study were vulnerable as characterized by collected demographic and clinical data. In focus groups and interviews, they shared their appreciation for being invited to participate in the study. They expressed feelings of severe fear, anxiety, and despair but, in the study, felt that someone cared about them, and they were no longer alone.

Results from the study may inform a new model of healthcare and community collaborations to reach underserved populations with variable health literacy competencies. Low-touch interventions developed using health literacy principles and delivered by a trusted partner—in our case, food assistance program drivers—are feasible to reach community-dwelling older adults during lockdown. This should be tested on a larger scale.

Despite efforts to disseminate COVID-19 information and resources via the internet—by the CDC and local health departments—very few seniors (at least in our study) used the internet regularly, and other methods of outreach are sorely needed. This study challenges the notion that usual public health mass communication is an effective strategy with older adults. Although most of the older adults in our study obtained their COVID information from the television news, the information left them confused and fearful.

Continued testing of new approaches is needed that facilitate ongoing connections with healthcare when the physical locations are unavailable, such as in a pandemic. Foregoing medical care, for acute as well as chronic care, has documented negative health outcomes. Alliance with community organizations that have access to individuals and their neighborhoods is a necessary and welcome collaboration that reinforces the importance of social determinants of health as well as medical care.

## Figures and Tables

**Table 1 healthcare-11-01856-t001:** Demographic and clinical descriptive statistics of study population at baseline.

	Total	Treatment	Control	*p*-Value ^1^
**Race/Ethnicity**				0.625 ^[f]^
Asian	3 (3.5%)	1 (1.2%)	2 (2.3%)	
Black	5 (5.8%)	4 (4.7%)	1 (1.2%)	
Caucasian/White	27 (31.0%)	11 (12.8%)	16 (18.6%)	
Latino/Hispanic	45 (51.7%)	21 (24.4%)	24 (27.9%)	
Other	7 (8.1%)	3 (3.5%)	4 (4.7%)	
**Preferred Language**				0.910
English	56 (64.4%)	26 (29.9%)	30 (34.5%)	
Spanish	31 (35.6%)	14 (16.1%)	17 (19.5%)	
**Education Completed**				0.615 ^[f]^
<8th Grade	28 (32.2%)	13 (14.9%)	15 (17.2%)	
Some High School	6 (6.9%)	4 (4.6%)	2 (2.3%)	
Completed HS	16 (18.4%)	7 (8.1%)	9 (10.3%)	
Some College	20 (23.0%)	10 (11.5%)	10 (11.5%)	
Bachelor’s Degree or higher	17 (19.5%)	1 (1.2%)	0 (0.0%)	
**Health Insurance**				0.130 ^[f]^
Employer	11 (12.6%)	2 (2.3%)	0 (0.0%)	
Medi-Cal	6 (6.9%)	4 (4.6%)	2 (2.3%)	
Medi-Medi	14 (16.1%)	7 (8.1%)	7 (8.1%)	
Medicare	53 (62.9%)	25 (28.7%)	28 (32.2%)	
No Insurance	3 (3.5%)	1 (1.2%)	2 (2.3%)	
**Living Situation**				0.154 ^[f]^
Alone	27 (31.0%)	10 (11.5%)	17 (19.5%)	
Other	3 (3.5%)	4 (4.6%)	5 (5.8%)	
Roommate	8 (9.0%)	1 (1.2%)	2 (2.3%)	
With Adult Child(ren)/family	19 (21.8%)	8 (9.0%)	10 (11.5%)	
With spouse/partner	30 (34.5%)	17 (19.5%)	13 (14.9%)	
**Age Categories**				0.644
60–64 yrs.	16 (18.4%)	7 (8.1%)	9 (10.3%)	
65–69 yrs.	18 (20.7%)	10 (11.5%)	8 (9.2%)	
70–74 yrs.	19 (21.8%)	6 (6.9%)	13 (14.9%)	
75–79 yrs.	18 (20.7%)	9 (10.3%)	9 (10.3%)	
80 yrs. or older	16 (18.4%)	8 (9.2%)	8 (9.2%)	
**Difficulty Paying for Medications**				0.613 ^[f]^
Always	2 (2.3%)	0 (0.0%)	2 (2.3%)	
Often	4 (4.6%)	2 (2.3%)	2 (2.3%)	
Sometimes	19 (21.8%)	10 (11.5%)	9 (10.3%)	
Rarely	10 (11.5%)	3 (3.5%)	7 (8.1%)	
Never	52 (59.8%)	25 (28.7%)	27 (31.0%)	
**Inadequate Health Literacy**	25 (28.7%)	12 (13.8%)	13 (14.9%)	0.810
**Inadequate and Marginal Health Literacy** (combined)	46 (52.9%)	20 (23.0%)	26 (29.9%)	0.620
**No Internet Access**	40 (46.0%)	21 (24.1%)	19 (21.8%)	0.260
**No Cell Phone**	11 (12.6%)	6 (6.9%)	5 (5.8%)	0.542
**Physical Single SF Question**				0.601 ^[f]^
Excellent	4 (4.6%)	3 (3.5%)	1 (1.2%)	
Very good	7 (8.1%)	3 (3.5%)	4 (4.6%)	
Good	30 (34.5%)	12 (13.8%)	18 (20.7%)	
Fair	40 (46.0%)	18 (20.7%)	22 (25.3%)	
Poor	6 (6.9%)	4 (4.6%)	2 (2.3%)	
**Prescription Medications**				0.108 ^[f]^
None	6 (6.9%)	0 (0.0%)	6 (7.0%)	
1–3 meds	29 (33.3%)	14 (16.3%)	14 (16.3%)	
4–6 meds	32 (36.8%)	13 (15.1%)	19 (22.1%)	
7–9 meds	14 (16.1%)	8 (9.3%)	6 (7.0%)	
10 or more meds	6 (6.9%)	4 (4.7%)	2 (2.3%)	
**Any Fall** (last 12 months)	33 (37.9%)	15 (17.2%)	18 (20.7%)	0.939
**Any ED** (last 12 months)	27 (31.0%)	14 (16.1%)	13 (14.9%)	0.461
**Any Hospitalization** (last 12 months)	19 (21.8%)	10 (11.5%)	9 (10.3%)	0.510
**Diabetes**	29 (33.3%)	16 (18.4%)	13 (14.9%)	0.224
**Hypertension**	48 (55.2%)	25 (28.7%)	23 (26.4%)	0.205
**COVID-19** (tested positive)	6 (6.9%)	5 (5.8%)	1 (1.2%)	0.090 ^[f]^
**Total Chronic Conditions**	2.8 [1.6; 0–7]	2.9 [1.4; 0–6]	2.6 [1.7; 0–7]	0.296
**Self-Efficacy Score**	9.1 [2.4; 2–12]	9.5 [2.3; 5–12]	8.8 [2.4; 2–12]	0.629

^1^ Categorical variables are reported as frequency (%) and continuous variables as mean [Standard Deviation; minimum to maximum values]; the *p*-values shown are the results from significance testing of the differences between treatment and control groups. Chi square test (or Fisher’s exact test indicated by ^[f]^) for small cell size) was used for significance testing of the categorical variables and independent samples *t*-test with the continuous variables.

**Table 2 healthcare-11-01856-t002:** Outcome measurements at baseline.

	Cohort	Treatment	Control	*p*-Value
COVID Mitigation Behavior Score (N)	87	40	47	0.024 *
Group ^1^	17.7 [2.9; 9–25]	18.5 [3.3; 9–25]	17.0 [2.3; 12–22]	
	Cohort	Treatment	Control	*p*-Value ^2^
COVID Knowledge	46 (52.9%)	18 (20.7%)	28 (32.2%)	0.175
No Telehealth	12 (13.8%)	5 (5.8%)	7 (8.1%)	1.000 ^[f]^
Exercise Frequency				0.117
None	24 (27.6%)	13 (14.9%)	11 (12.6%)	
1 day	9 (10.3%)	1 (1.2%)	8 (9.2%)	
2 to 3 days	18 (20.7%)	8 (9.2%)	10 (11.5%)	
4 days	3 (3.5%)	1 (1.2%)	2 (2.3%)	
5 or more days	33 (37.9%)	17 (19.5%)	16 (18.4%)	
Loneliness Risk	31 (35.6%)	14 (16.1%)	17 (19.5%)	0.910

* Independent samples *t*-test used for significance testing of the difference in means between treatment and control groups; significant at the <0.05 level. ^1^ The score was treated as a continuous variable and reported with respective mean [standard deviation; minimum to maximum values]. ^2^ Outcomes reported as frequency (%); Chi square test (or Fisher’s exact test indicated by ^[f]^ for small cell size) was used for significance testing of the differences between treatment and control groups at baseline. No differences were observed for these outcomes.

**Table 3 healthcare-11-01856-t003:** Results—difference-in-differences estimates for study outcomes.

Primary Outcome	FE Estimate	DID Estimate	SE	*t* Value	Pr > |*t*|
COVID Mitigation Score					
**Post * Treatment**		**−1.563**	**0.756**	**−2.07**	**0.042 ***
Post	3.213		0.512	6.27	<0.0001 **
Treatment	1.338		0.544	2.45	0.016 *
Any ED	1.140		0.499	2.28	0.025 *
Any IP	0.332		0.545	0.61	0.543
Inadequate HL	0.066		0.494	0.13	0.894
Spanish language	1.182		0.455	2.60	0.011 *
Restricted Maximum Likelihood Model Estimation					
	**GEE SE Estimate**	**DID Estimate**	**SE**	**Z Value**	**Pr > |Z|**
COVID Knowledge ^1^					
Post * Treatment		0.036 (CI: −0.249–3.20)	0.145	0.25	0.803
Post	0.064		0.140	0.73	0.464
Treatment	−0.146		0.106	−1.37	0.171
**Secondary Outcome**					
Exercise (frequency) ^2^					
Post * Treatment		−0.355 (CI: −1.018–0.307)	0.338	−1.05 −1.05	0.292
Post	0.698	(CI: 0.334–1.0630)	0.186	3.76	0.0002 **
Treatment	0.121		0398	0.30	0.761
Loneliness Index ^3^					
Post * Treatment		0.228 (CI: −0.572–0.027)	0.408	0.56	0.577
Post	−0.008		0.280	−0.03	0.979
Treatment	−0.594	(CI: −1.312–0.124)	0.366	−1.62	0.105
No Telehealth Visit ^4^					
Post * Treatment		−0.007 (CI: −0.096–0.081)	0.045	−0.16	0.870
Post	−0.043		0.030	−1.45	0.148
Treatment	−0.024		0.074	−0.16	0.870

Note: ** *p* < 0.01, * *p* < 0.05. ^1.^ Generalized Estimating Equations with binomial distribution and identity link used for estimating a population-level marginal D-I-D model. ^2 & 3.^ Generalized Estimating Equations with multinomial distribution and cumulative log link for estimating a population-level, marginal D-I-D model. ^4.^ Generalized Estimating Equations with binomial distribution and identity link for estimating a population-level marginal D-I-D model.

## Data Availability

The data presented in this study are available on request from the corresponding author. The data are not publicly available as additional publications of subsample populations are in process.

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
