# Peer review of "Health Support for At-Risk Older Adults during COVID-19"

_healthcare, 2023, doi:10.3390/healthcare11131856_

Round 1
Reviewer 1 Report
This study aimed to investigate the feasibility of the new ‘low-touch’ intervention approach, a health literacy best practice and its impact on COVID-19 knowledge, mitigation behaviors, telehealth visits, self-care, loneliness, and fall prevention activities.
Title
Clear
Abstract
Clear
Introduction
The introduction related to the current literature and explained the research gap. It also justified the need for this study. The aims of study were well presented and clear.
Material and methods
Should add 2.3 for data collection to describe the questions involved in different aspects and also how to assess such as yes or no or a 5-point Likert scale.
If all the questions in the survey were tested and validated in older adults over phone before the conduction of this survey? If so, how many older adults were tested?
If the surveys translated into Spanish were also validated as more than one-third of the surveys were conducted in Spanish, how to make sure the Spanish version would get similar response as English version and the data can be analysed without bias?
For session 2.5, the descriptions on how to assess COVID mitigation behavior and knowledge, loneliness and self-efficacy etc should be mentioned in the section for data collection.
Outcome measures should define the primary and secondary outcomes and how to evaluate them.
How the authors designed the questions related to COVID-19 mitigation and knowledge? No reference can be provided here.
Sorry, I did not quite understand line 232-234 on the significant differences here.
Result
The authors should provide the reasons why 11 of the participants did not complete the survey.
Table 1 should also report the demographic background and clinical variables in both groups for comparison
Discussion
The authors should discuss more about the results.
Limitation: the recruited participants were limited to a specific cohort that required food assistance. The proportion of Latino was over 50% which may not be the same as USA population. The result may not be generalized.
Conclusion
The conclusion should be more precise. It should address the objectives listed before and concluded from the result findings.
Author Response
Response to Reviewer 1 Comments
[Material and methods]
Point 1: Should add 2.3 for data collection to describe the questions involved in different aspects and also how to assess such as yes or no or a 5-point Likert scale.
Response 1:
Section 2.3 has been added to the revised manuscript to better describe the construction of the COVID-19 Mitigation Behavior composite (summation) score and the COVID-19 Knowledge questions as well as all other outcome variables.
Revised copy for new section:
2.3 Data Collection
Data was primarily collected from participants’ completed baseline and post study surveys. The survey was developed and iterated following internal, informal field testing by researchers. Many of the questions were adapted from questions in Medicare and Medi-Medi Health Risk Assessment surveys used in the County. Medical history and clinical questions were included to establish a study cohort risk profile and identify at-risk older adults in need of health care and services. As previously discussed, research assistants were trained in the administration of the survey and in using scripted protocols for questions and prompts if needed. Two of the four research assistants were proficient in both English and Spanish and they were assigned all study participants preferring Spanish language. All the resources in the senior kit had been translated into Spanish, field tested, and revised as needed prior to the study. Most of the questions in the survey came from other validated surveys or surveys in current use by health plans that are available in Spanish, English, and a variety of other languages and are administered telephonically.
The key outcome variables were COVID-19 mitigation behavior and COVID-19 knowledge. These questions were developed by aligning with stay-at-home state and public health orders. The first three COVID-19 mitigation behaviors included the frequency of leaving the home (older adults were to stay at home as much as possible), visits by groups of friends/families (older adults were not to have groups of people in their homes), attending large gatherings (no one was supposed to attend large gatherings) – response options included never to weekly (scored 5 to 1; reverse scored as these are undesirable behaviors). Wearing masks (first mandate of the public health order) when going outside was measured with response options of never to always (scored 1 to 5), and the likelihood of going without or delaying urgent care because of COVID-19 was measured with response options of very unlikely to very likely (scored 1 to 5). Early in the pandemic many physicians were concerned that older adults may be avoiding the ED in serious situations due to fear; this has been confirmed[39, 40]. Summing the scores for each question led to composite score, with a range of 5 to 25.
COVID-19 knowledge was assessed by three questions believed by the researchers to be important to keeping oneself safe: 1) Can people have COVID-19 and not know it? 2) Is it safe to go to an event where many people are present, as long as, they are 6 feet apart? 3) Is it safe to go to the hospital if you are really sick? One point was assigned for each correct response yielding a COVID knowledge composite score of 0 to 3.
Other study outcomes included improving access to telehealth, increasing exercise, and decreasing loneliness. Access to telehealth visits was assessed by two survey questions: 1) In the last 6 months, did you have a telehealth visit with your doctor? 2) In the last 6 months, did you see your doctor in-person? The post survey timeframe for these questions were modified to since we spoke last, did you…These questions were converted to a single binary variable (1/0) with a 1 assigned if they had a visit with their doctor.
Exercise frequency was measured at baseline and post study with a single question, On how many days a week are doing some kind of exercise now? Response options included: No days, 1 day, 2 to 3 days, 4 days, 5 days or more; scored as 0, 1, 2, 3, 4 respectively.
Loneliness was measured using the shortened version of the UCLA Loneliness Index [26]. The survey consists of three questions: 1) How often do you not have someone to talk with? 2) How often do you feel left out? 3) How often do you feel disconnected from others? (Response options for all three questions are: hardly ever, some of the time, often, scored as 1 to 3). The index ranges from 3 to 9; greater than or equal to 6 indicates high risk for loneliness – this short version received an acceptable reliability score of 0.72.
Point 2: If all the questions in the survey were tested and validated in older adults over phone before the conduction of this survey? If so, how many older adults were tested?
Response 2: The questions taken from validated surveys had been used also in Spanish. The Health Risk Assessment (HRA) survey is commonly used in our County for all Medicare and Medi-Medi beneficiaries, so these questions and their format are familiar to many older residents – required to be written at the 6th grade reading level or below. Both organizations conducted informal internal field testing of the survey with several seniors. Drs. Gibbs and Sehgal provide care at Senior Health Centers and my organization works with at-risk seniors and has senior employees. The survey was revised about 5 times before protocols were written. This information is contained now in the opening of the new 2.3 Data Collection section (above).
Point 3: If the surveys translated into Spanish were also validated as more than one-third of the surveys were conducted in Spanish, how to make sure the Spanish version would get similar response as English version and the data can be analysed without bias?
Response 2: Although not a guarantee, administering surveys over the phone using standardized protocols inclusive of prompts for questions is a best practice for data collection by survey.
Point 4: For session 2.5, the descriptions on how to assess COVID mitigation behavior and knowledge, loneliness and self-efficacy etc. should be mentioned in the section for data collection. Outcome measures should define the primary and secondary outcomes and how to evaluate them.
Response 4: This portion of 2.5 has been moved to the new 2.3 data collection per your suggestion. Thank you. Self-efficacy has been removed as a study outcome. It was not an outcome in the original study design and was collected as a composite of 3 validated self-efficacy questions as part of the study participant baseline profile. As it can influence behavior if differences were observed in self-efficacy between the study groups it could be controlled for. There was no observed difference in self-efficacy, however between treatment and control groups.
Section 2.5 has been revised per copy below.
2.5. Outcome Measures
Primary outcome measures were COVID-19 mitigation behaviors and COVID-19 knowledge. The primary goals of this study were to improve both mitigation behavior and COVID knowledge to keep the older residents as safe as possible.
The COVID-19 mitigation behavior score was assessed for differences between treatment and control groups as an interval composite variable with the higher the score the more the person was fully complying with recommended public health mitigation behaviors. Given the sample size and scale range of this variable a parametric test for this outcome measure may be used [31, 32]. In the final analysis a mixed linear model with repeated measures of treatment and time on the COVID mitigation behaviors score with a random intercept for pre/post measures by treatment group is estimated. Generalized linear mixed models do not assume your dependent variable is normally distributed.
COVID-19 knowledge was collected with three questions. For analysis, this variable was converted to a binary variable (1/0) with a 1 assigned if all three questions were answered correctly. As it was expected that COVID-19 knowledge would impact behaviors it was important to determine baseline and post COVID knowledge.
Secondary outcome measures included loneliness, telehealth access, and exercise. The 3-Item Loneliness Index was converted to a binary variable (1/0) with a 1 assigned for at-risk for loneliness if the total score was 6 or more for the analysis. The 3-Item Loneliness Index represents a screener for identifying people with high levels of loneliness for whom interventions would be appropriate. This study engaged with a group of older adults sheltering at home and often with little to no contact with others. It was believed participation in the study would decrease their risk for loneliness.
Exercise frequency was used to examine changes in exercise during the study. At-home exercise video (and private URL) was included in the senior kit and participants were encouraged to do some of these every day. This component was designed (evidence-based exercises demonstrated on the video) to prevent avoidable falls in the home and maintain function during this time of staying at home. Exercise frequency was analyzed as an ordered, multinomial variable.
Point 5: How the authors designed the questions related to COVID-19 mitigation and knowledge? No reference can be provided here.
Response 5: This has been addressed in the new data collection copy, 2.3 previously inserted. There were few references at the time – the pandemic had just hit, and this study was developed and submitted the end of May 2020. The questions were developed from state stay-at-home and public health orders.
Point 6: Sorry, I did not quite understand line 232-234 on the significant differences here.
Response 6: Thank you – this was poorly explained. The copy has been revised to the following.
A Difference-in-Differences method was used to analyze changes in pre- and post-study outcomes between treatment and control groups. The benefit of such a model is that it accounts for secular changes occurring in the environment exposing both study groups to parallel trends across time. In the case of this study, the cohort was fixed controlling for measured and unmeasured confounding. Generalized linear models were used to generate the “difference-in-differences” estimates for all outcome measures using the appropriate model specification for the variable type (continuous, binary, or ordinal). A difference-in differences (D-I-D) estimate is the difference between the difference in treatment post and pre-parameter estimates, and the difference in control post and pre-parameter estimates. Sometimes referred to as a controlled before and after study as participants serve as their own controls. The D-I-D estimates the between-group cross-sectional differences and the within-group time series differences to measure treatment effects [33-37]. We included the dummy variable for inadequate health literacy on outcomes if it improved the model as it is the key interest in this study. All significance testing was set at < 0.05. All quantitative analyses were performed using SAS proprietary software 9.4 and analytical products SAS/STAT 15.1 (SAS Institute, Inc., Cary, NC).
Result
Point 7: The authors should provide the reasons why 11 of the participants did not complete the survey.
Response 7: The 11 non-completers were those who could not be reached at the four-month point (the end of the study). Bivariate analyzes were done on all baseline parameters between the 11 who were lost to follow-up and the 87 who completed both the baseline and post-study surveys and no differences were observed. As this study was conducted early in the pandemic, we could not visit study participants to collect end of study data if they did not respond to outreach phone calls. We called five times and mailed two reminders.
Point 8: Table 1 should also report the demographic background and clinical variables in both groups for comparison
Response 8: Table 1 has been revised to include all variables reported in total and by study group. The revised table is included at the end of this review.
Discussion
Point 9: The authors should discuss more about the results.
Response 9: Thank you – this section has been revised.
Revised copy for this section.
- Discussion
In this feasibility study of community-dwelling older adults residing under stay-at-home orders, a “low touch” intervention, a Senior Resource Box with engaging, easy-to-read materials on staying safe and healthy during COVID-19 improved mitigation behaviors recommended by public health, and the state and local authorities (<.0001). Study participants were randomized into control and treatment groups; however, the entire cohort received the Senior Resource Box as the “low touch” intervention, and the treatment group additionally received health coaching support calls. Contrary to our hypothesis, the health coaching support calls did not enhance this “low touch” intervention (written materials and DVD) and increase our mitigation behavior score beyond that observed in the control group. The difference-in-differences estimate was negative (-1.563) and significant (<0.42). Although both study groups increased their COVID mitigation behavior scores, the difference between each study group’s mean score difference (post-pre) was larger for the control group, not the treatment group. Notably, the control group’s least squares mean mitigation behavior score at baseline (17.6) was much lower as compared with the treatment group (18.9). The post mitigation behavior scores were closer at the end of the study, but the control group was slightly higher (20.8 versus 20.5). No differences were found when comparing the demographic and clinical data between treatment and control groups – even for variables one might anticipate influencing COVID mitigation behavior.
This unexpected finding could be that the health support calls focused on other health concerns – the study protocol was for the health coach to first attend to any risk responses from the participant at baseline. Another possibility is the plain language materials regarding COVID-19 and how to keep oneself safe improved any confusion an older adult may have had about public health recommendations [42]. As these materials were engaging, easy-to-read and understandable, and delivered by a trusted person the strength of the Senior Resource Box alone increased the mitigation behavior of both groups [43, 44].
There was no observed change in the difference-in-differences estimate for COVID-19 knowledge and no increase across the cohort. The COVID-19 knowledge variable was assessed as a binary variable – responding to all questions correctly because the information contained in these questions were required to follow mitigation behaviors. The non-significant negative estimate for treatment (health support calls) may suggest COVID-19 knowledge was not a focus of the calls.
There were no significant findings on our secondary outcomes for the difference-in-differences estimate in support of our hypothesis that the difference between the treatment (post-pre rates) and control (post-pre rates) would be positive for increase exercise, decrease loneliness, and decrease the number of older adults without a telehealth or other doctor contact. Very unexpectedly, only one of our secondary outcomes had any significant improvement across the cohort over the post period. Increase in exercise to preserve physical function and support physical conditioning over the post period of the study was significant (p=0.0002).
Insignificant findings for decreasing loneliness across the cohort may be the result of the chosen screening instrument for a “lock-down” situation. The instrument was chosen as it was brief, specific to loneliness, and had been validated. It was expected that the health support calls would decrease loneliness through these regular contacts. The non-significant treatment parameter estimate (-0.594, p=0.105) may suggest the calls did decrease loneliness of the older adults in the treatment group. The insignificant finding in the lack of telehealth (or other doctor contact) outcome was not surprising. Only 13.8% of the study cohort (12 older adults) had not had a telehealth visit or other contact with their doctor at baseline. These baseline results were unexpected as this was an expressed concern from physicians. At the end of the study four of these 12 older adults did have a visit with their doctors, a clinically, although statistically non-significant meaningful result.
Studies of effective health communication strategies in an emergent situation are limited [43, 45]. The last time an urgent public health emergency played out widely was over one hundred years ago, but the few studies shown indicate a trusted messenger is critical and plain language information had delivered to residents is needed to meet the needs of vulnerable populations [43-45].
Our qualitative findings from focus groups and interviews were consistent with research during the timeframe [46-48]. The COVID-19 stay-at-home orders were challenging times for older adults and innovative ways to communicate with them is critical [49]. It resulted in a negative impact on their health, mental health, and physical health and the resulting social isolation and decreased physical activity further threatened their health [50].
This study had several limitations, but it was intended as a pilot study to test a new approach to get critical health information and support to older adults during the pandemic lockdown. The study population was restricted to a single location although one representative of older adult risk in many other areas – culturally diverse, eligible for the senior food assistance program, and of moderate clinical risk. The COVID-19 knowledge and mitigation behavior questions were limited to minimize survey burden and the desire to gather medical risk data to accurately profile this community population. The engagement rate was much lower than anticipated at 47% and we do not know anything about those older adults who received the Senior Health Resource Box (the “low touch” intervention) but failed to join the study. The study was conducted during a pandemic which included challenges for older adults, especially in underrepresented communities who were the hardest hit [51]. The chaos during this time may have overshadowed those who would have participated in the study otherwise. Also, the survey itself was comprised of questions from multiple validated surveys, however the overall survey was not validated.
The failure to find significant differences in many of the outcome measures may be related to the short length of follow-up (four months) and the failure to recruit as many participants as intended during the pandemic.
Point 10: Limitation: the recruited participants were limited to a specific cohort that required food assistance. The proportion of Latino was over 50% which may not be the same as USA population. The result may not be generalized.
Response 10: This is a true statement. The findings of this study are not meant to be generalizable but rather be used to seek funding to further explore the important public and health policy related to finding effective strategies to support our most vulnerable residents during our next pandemic or public health emergency. These older adults were of lower household income (by nature of eligibility for the senior food assistance program) and of moderate clinical risk as shown in the study profile. I did frame the study population as being residents of La Habra in southern California – in this city Latinos are the majority population as is true in many cities in California and other parts of the U.S. They are an important population for further study and finding effective ways for public health outreach. Not every person needed the same level of support during COVID-19 but if more vulnerable people had been reached lives could have been saved. This study was about testing potential strategies that could be used to fuel thought and more funded research in this important area.
Conclusion
Point 11: The conclusion should be more precise. It should address the objectives listed before and concluded from the result findings.
Response 11: The conclusion has been revised.
Revised copy.
- Conclusions
Although this study was conducted as a feasibility study, it has important implications. The study did demonstrate the feasibility of approach ─ the engagement of older adults through a trusted community partner (in our case, senior food program drivers), and delivery of a “low touch” intervention with accessible health information from a trusted person. Although we failed to find a positive impact from the enhancement of health support calls (using a difference-in-differences estimation), we did observe an increase in COVID-19 mitigation behavior across the overall cohort (<.0001). We also observed an increase in exercise across the study cohort (p=.0002) that has the potential to reduce avoidable falls, improve immune response, and promote feelings of well-being among older adults.
A trusted community partner was an essential implementation factor, but the timeline for recruitment and the study duration should have been extended. The older adults enrolled in the study were vulnerable as characterized by collected demographic and clinical data, but in focus groups and interviews they shared their appreciation for being invited to participate in the study. They expressed feelings of severe fear, anxiety, and despair, but in the study felt someone cared about them and they were no longer alone.
Results from the study may inform a new model of healthcare and community collaborations to reach underserved populations with variable health literacy competencies.
Despite efforts to disseminate COVID-19 information and resources via the Internet – by CDC and local health departments, very few seniors (at least in our study) used the Internet regularly (8.3%), and other methods of outreach are sorely needed.
Although failing to show a positive difference in outcomes between the treatment group’s post-pre measures compared with the control group’s post-pre measures, the health coaches helped with setting up telehealth visits, facilitating prescription medication receipt, encouraging at-home exercise, and connecting them with classes to learn how to use their smartphones as a computer and maintaining bonds with loved ones.
Low-touch interventions developed using health literacy principles are feasible within a community setting of older adults during lockdown that can be reached through a trusted partner – in our case food assistance program agents. These should be tested on a large scale.
This study challenges the notion that usual public health mass communication is an effective strategy with older adults. Although most of the older adults in our study obtained their COVID information from the television news, the information left them confused and fearful. New approaches are needed that facilitate ongoing connections with healthcare when the physical locations are unavailable, such as in a pandemic. Foregoing medical care, for acute as well as chronic care, has documented negative health outcomes. Alliance with community organizations that have access to individuals and their neighborhoods is a necessary and welcome collaboration that reinforces the importance of social determinants of health as well as medical care.
Table 1 Revision (attachment for Response 8).
Table 1. Demographic and Clinical Descriptive Statistics of Study Population at Baseline
|
|
|
|
|
|
Total |
Treatment |
Control |
p-Value1 |
Race/Ethnicity |
|
|
|
0.625[f] |
Asian |
3 (3.5%) |
1 (1.2%) |
2 (2.3%) |
|
Black |
5 (5.8%) |
4 (4.7%) |
1 (1.2%) |
|
Caucasian/White |
27 (31.0%) |
11 (12.8%) |
16 (18.6%) |
|
Latino/Hispanic |
45 (51.7%) |
21 (24.4%) |
24 (27.9%) |
|
Other |
7 (8.1%) |
3 (3.5%) |
4 (4.7%) |
|
Preferred Language |
|
|
|
0.910 |
English |
56 (64.4%) |
26 (29.9%) |
30 (34.5%) |
|
Spanish |
31 (35.6%) |
14 (16.1%) |
17 (19.5%) |
|
Education Completed |
|
|
|
0.615[f] |
< 8th Grade |
28 (32.2%) |
13 (14.9%) |
15 (17.2%) |
|
Some High School |
6 (6.9%) |
4 (4.6%) |
2 (2.3%) |
|
Completed HS |
16 (18.4%) |
7 (8.1%) |
9 (10.3%) |
|
Some College |
20 (23.0%) |
10 (11.5%) |
10 (11.5%) |
|
Bachelor’s Degree or higher |
17 (19.5%) |
1 (1.2%) |
0 (0.0%) |
|
Health Insurance |
|
|
|
0.130[f] |
Employer |
11 (12.6%) |
2 (2.3%) |
0 (0.0%) |
|
Medi-Cal |
6 (6.9%) |
4 (4.6%) |
2 (2.3%) |
|
Medi-Medi |
14 (16.1%) |
7 (8.1%) |
7 (8.1%) |
|
Medicare |
53 (62.9%) |
25 (28.7%) |
28 (32.2%) |
|
No Insurance |
3 (3.5%) |
1 (1.2%) |
2 (2.3%) |
|
Living Situation |
|
|
|
0.154[f] |
Alone |
27 (31.0%) |
10 (11.5%) |
17 (19.5%) |
|
Other |
3 (3.5%) |
4 (4.6%) |
5 (5.8%) |
|
Roommate |
8 (9.0%) |
1 (1.2%) |
2 (2.3%) |
|
With Adult Child(ren)/family |
19 (21.8%) |
8 (9.0%) |
10 (11.5%) |
|
With spouse/partner |
30 (34.5%) |
17 (19.5%) |
13 (14.9%) |
|
Age Categories |
|
|
|
0.644 |
60-64 yrs. |
16 (18.4%) |
7 (8.1%) |
9 (10.3%) |
|
65-69 yrs. |
18 (20.7%) |
10 (11.5%) |
8 (9.2%) |
|
70-74 yrs. |
19 (21.8%) |
6 (6.9%) |
13 (14.9%) |
|
75-79 yrs. |
18 (20.7%) |
9 (10.3%) |
9 (10.3%) |
|
80 yrs. or older |
16 (18.4%) |
8 (9.2%) |
8 (9.2%) |
|
Difficulty Paying for Medications |
|
|
|
0.613[f] |
Always |
2 (2.3%) |
0 (0.0%) |
2 (2.3%) |
|
Often |
4 (4.6%) |
2 (2.3%) |
2 (2.3%) |
|
Sometimes |
19 (21.8%) |
10 (11.5%) |
9 (10.3%) |
|
Rarely |
10 (11.5%) |
3 (3.5%) |
7 (8.1%) |
|
Never |
52 (59.8%) |
25 (28.7%) |
27 (31.0%) |
|
Inadequate Health Literacy |
25 (28.7%) |
12 (13.8%) |
13 (14.9%) |
0.810 |
Inadequate and Marginal Health Literacy (combined) |
46 (52.9%) |
20 (23.0%) |
26 (29.9%) |
0.620 |
No Internet Access |
40 (46.0%) |
21 (24.1%) |
19 (21.8%) |
0.260 |
No Cell Phone |
11 (12.6%) |
6 (6.9%) |
5 (5.8%) |
0.542 |
Physical Single SF Question |
|
|
|
0.601[f] |
Excellent |
4 (4.6%) |
3 (3.5%) |
1 (1.2%) |
|
Very good |
7 (8.1%) |
3 (3.5%) |
4 (4.6%) |
|
Good |
30 (34.5%) |
12 (13.8%) |
18 (20.7%) |
|
Fair |
40 (46.0%) |
18 (20.7%) |
22 (25.3%) |
|
Poor |
6 (6.9%) |
4 (4.6%) |
2 (2.3%) |
|
Prescription Medications |
|
|
|
0.108[f] |
None |
6 (6.9%) |
0 (0.0%) |
6 (7.0%) |
|
1-3 meds |
29 (33.3%) |
14 (16.3%) |
14 (16.3%) |
|
4-6 meds |
32 (36.8%) |
13 (15.1%) |
19 (22.1%) |
|
7-9 meds |
14 (16.1%) |
8 (9.3%) |
6 (7.0%) |
|
10 or more meds |
6 (6.9%) |
4 (4.7%) |
2 (2.3%) |
|
Any Fall (last 12 months) |
33 (37.9%) |
15 (17.2%) |
18 (20.7%) |
0.939 |
Any ED (last 12months) |
27 (31.0%) |
14 (16.1%) |
13 (14.9%) |
0.461 |
Any Hospitalization (last 12 months) |
19 (21.8%) |
10 (11.5%) |
9 (10.3%) |
0.510 |
Diabetes |
29 (33.3%) |
16 (18.4%) |
13 (14.9%) |
0.224 |
Hypertension |
48 (55.2%) |
25 (28.7%) |
23 (26.4%) |
0.205 |
COVID-19 (tested positive) |
6 (6.9%) |
5 (5.8%) |
1 (1.2%) |
0.090[f] |
Total Chronic Conditions |
2.8 [1.6; 0-7] |
2.9 [1.4; 0-6] |
2.6 [1.7; 0-7] |
0.296 |
Self-Efficacy Score |
9.1 [2.4; 2-12] |
9.5 [2.3; 5-12] |
8.8 [2.4; 2-12] |
0.629 |
1 Categorical variables are reported as frequency (%) and continuous variables as mean [Standard Deviation; minimum to maximum values]; p-values shown are the results from significant testing of the differences between treatment and control groups. Chi Square test (or Fisher’s Exact Test indicated by [f]) for small cell size) was used for significant testing of the categorical variables and independent samples t-test with the continuous variables.
Reviewer 2 Report
The article entitled:
Health Support for At-Risk Older Adults During COVID-19.
The text discusses covid-19 infection in the elderly population.
It makes a contextualization of covid, and evaluates the effect of telehealth in control and intervention groups.
The study is interesting to know in the elderly population and provides valuable content to the specific research area. It also describes in great detail how you did the study which is to be appreciated, however, I have 2 recommendations to make.
First, improve the introduction, I found it scarce, brief and with little depth. Expand in quotes your conceptualization, extend more and conceptualize the state of the art.
On the other hand, to increase the internationalization of your study, downloads and citations I recommend you to incorporate: "The practice of physical activity is one of the variables that we should not forget in this area, as it improves the quality of life of the elderly as well as a guarantee of good aging in terms of health, and that even aspects such as gender, educational level, health, functional skills and leisure activities are socio-contextual elements that determine health care in the elderly population and that influence their quality of life" https://doi. org/10.3390/bs12090331 "Likewise, socio-environmental and contextual factors related to a higher quality of life, also influence a greater practice of physical activity in older people such as income and education, being training on healthy lifestyle habits one of the axes to acquire healthy lifestyle habits, such as practicing greater physical activity in older people, so, better understand the motives and variables that influence the practice of physical activity in older people" https://doi.org/10.3390/ijerph182010815
And my second recommendation focuses on the discussion. It is well debated, well contrasted, but at the end, I found some questions missing, which I consider interesting to reflect on:
-Indicate what implications this work has for the rest of scientific scholars studying this topic.
-What are the practical implications for the elderly?
-Any interesting future lines that need to be covered in the research.
I hope you get better visibility this way! Congratulations, I loved your work!
My sincere congratulations for the work.
Reviewer 3 Report
This study focuses on the efficacy of an health coach assisted older adults in their knowledge and health behaviors related to COVID as well as a number of secondary characteristics. I have several observations about this paper:
1. most importantly, the analyses conducted are presented in a very confusing manner-the tables reflect a mix of parametric and nonparametric methods and frankly, are very difficult to discern given the disorganized manner in which the findings are analyzed and presented. Amid the confusing manner in which data are analyzed and findings presented, it appears that the health coach enhanced program was not effective.
2. The primary outcome measures should have been analyzed via a 2 x 2 (M)ANOVA. with post hoc comparisons performed contingent upon a group by time interaction. The remainder of the findings seem supplementary and less important.
3. The apparently negligible program impact could be an artifact of unreliable outcome measures (reliabilities for the primary outcome measures are not presented) and/or attrition. This should be explored.
4. The intervention should be described first, followed by the control condition. It should be made explicitly clear that persons were randomly assigned to treatment and control conditions.
5. Relying upon multiple questions to assess efficacy via 3 questions pre-program and relying on a single question to do so is unacceptable.
6. The description of the sample belongs in the methods section not results.
7. The authors might explore whether treatment/control effects over time appear when crossing levels of health literacy across participants-thus there is the potential for a 3 way interaction.
Round 2
Reviewer 1 Report
Thank you for the update and revision. I have no further comments.
Author Response
Reviewer 1 had no further comments for our team to address.
Reviewer 3 Report
I understand your commitment to your analytic strategy-while I still believe a MANOVA is a more straightforward manner of analysis, spelling out how you design satisfies the 4 criteria for the D-I-D approach should be adequate.
Author Response
Response: We have added copy, lines 402 to 418 at the end of Statistical Analysis Section 2.7 to support the minimum criteria and 4 assumptions required for a D-I-D approach.
Manuscript revision:
Page 12. “Importantly, the minimum requirements and key assumptions for a D-I-D model were satisfied. D-I-D approach requires the collection of outcome data on an exposed or treated group and a group not exposed to the intervention (control) during at least one time period before the exposure or intervention and at least one time period after the intervention. The allocation of treatment cannot be determined by the outcome; our intervention was by study design and not allocated by the outcome. The treatment and control groups can be assumed to have parallel trends in outcome. As only baseline and post-4-month measurements were collected, the parallel trend prior to the intervention cannot be directly measured. However, the short timeframe between pre-intervention and post-intervention data collection and the randomization of the participants to treatment and control, support this assumption holds. The composition of the treatment and control groups was stable over time and no differences were observed at baseline between the groups, nor were any differences found between the 87 older adults completing the study and the 11 study participants who could not be contacted for post-data collection. Repeat measures were collected on the same individuals at both time periods and the mixed procedure model with random effects accounted for the correlation between these measures for the same individual. There were no known spillover effects during the study. This study took place at the height of the COVID-19 pandemic and study participants in the treatment and control groups lived in the same city and were exposed to the same public health orders, COVID-19 infection rates, and access to preventive COVID-19 education and resources. Additionally, simple random assignment procedures kept two older adults living in the same household in the same group; older adults were assigned to the random assignment of the first person in the household.”